# Does weight management research for adults with severe obesity represent them? Analysis of systematic review data

Clare Robertson ![ORCID], Magaly Aceves-Martins ![ORCID], Moira Cruickshank ![ORCID], Mari Imamura ![ORCID], Alison Avenell ![ORCID]

Health Services Research Unit, University of Aberdeen, Aberdeen, UK

**Correspondence to**
Clare Robertson;
c.robertson@abdn.ac.uk

## ABSTRACT

**Objective** Our objective was to determine the extent to which current evidence from long-term randomised controlled trials (RCTs) of weight management is generalisable and applicable to underserved adult groups with obesity (body mass index (BMI) ≥35 kg/m$^2$).

**Methods** Descriptive analysis of 131 RCTs, published after 1990–May 2017 with ≥1 year of follow-up, included in a systematic review of long-term weight management interventions for adults with BMI ≥35 kg/m$^2$ (the REBALANCE Project). Studies were identified from MEDLINE, EMBASE, PsychINFO, SCI, CENTRAL and from hand searching. Reporting of trial inclusion and exclusion criteria, trial recruitment strategies, baseline characteristics and outcomes were analysed using a predefined list of characteristics informed by the PROGRESS (Place of residence, Race/ethnicity/culture/language, Occupation, Gender/sex, Religion, Education, Socioeconomic status, Social capital)-Plus framework and the UK Equality Act 2010.

**Results** Few (6.1%) trials reported adapting recruitment to appeal to underserved groups. 10.0% reported culturally adapting their trial materials. Only 6.1% of trials gave any justification for their exclusion criteria, yet over half excluded participation for age or mental health reasons. Just over half (58%) of the trials reported participants' race or ethnicity, and one-fifth reported socioeconomic status. Where outcomes were reported for underserved groups, the most common analysis was by sex (47.3%), followed by race or ethnicity (16.8%). 3.1% of trials reported outcomes according to socioeconomic status.

**Discussion** Although we were limited by poor trial reporting, our results indicate inadequate representation of people most at risk of obesity. Guidance for considering underserved groups may improve the appropriateness of research and inform greater engagement with health and social care services.

**Funding** National Institute for Health Research Health Technology Assessment Programme (project number: 15/09/04).

**PROSPERO registration number** CRD42016040190.

## STRENGTHS AND LIMITATIONS OF THIS STUDY

⇒ A unique analysis exploring whether randomised trials evaluating interventions for weight management for people with severe obesity consider the needs of underserved groups.
⇒ Data set includes up-to-date, best available randomised controlled trial evidence (considered to be the highest level of evidence to inform guidelines and clinical practice) identified using robust and exhaustive literature search strategies.
⇒ Data set includes wide-ranging weight management interventions delivered in different settings, from an international perspective.
⇒ Our analysis was limited by poor reporting of whether subgroup reporting by underserved groups was planned and whether trials were adequately powered to detect subgroup differences.
⇒ Non-randomised study designs, interventions with shorter follow-up and unpublished studies may have been more inclusive in their designs and reporting.

incomes, less education, lower socioeconomic status (SES) and disability are associated with greater risk of obesity for adults.[1–5] While the underlying causes of obesity are varied, there is an increasing association between obesity and deprivation that is driving poorer health outcomes and increasing health inequalities.[6] For example, most of these risk factors are stronger for women than men,[1–3 7] although men with obesity may be less likely than women to undertake weight management programmes.[8] Being an adult with obesity is associated with a lower health-related quality of life than a healthy adult of the same SES.[9] More severe obesity, such as body mass index (BMI) ≥35 kg/m$^2$ or more, with its associated greater risks for comorbidities, reduced quality of life and premature mortality,[10] is particularly related to lower SES,[1 11] and intellectual and physical disabilities.[4] Poorer outcomes from COVID-19 are also strongly

## BACKGROUND

In high-income countries, and increasingly in low/middle-income countries, lower

related to obesity, particularly severe obesity.[5] For countries such as the USA, UK, Australia and New Zealand, some racial or ethnic groups may also be at much greater risk of obesity, especially severe obesity.[11–14]

Preventing obesity and providing effective interventions, particularly for people with more severe obesity, are, therefore, a major public health challenge and vital in terms of addressing health inequalities. While organisations such as the US Food and Drug Administration,[15–18] the National Institutes of Health (NIH)[15] and the National Institute for Health Research (NIHR)[19] have produced guidelines on the inclusion of individuals of all ages, sexes/genders, races and ethnicities, and other physical, sensory/perceptual, cognitive and emotional characteristics, there is a lack of accessible policy-ready evidence on what works in terms of interventions to reduce inequalities in obesity. It is also recognised that some groups (for example, socially disadvantaged, less educated, and minority race or ethnic groups) may be less likely to be recruited into randomised controlled trials (RCTs) for lifestyle interventions.[20–24] Similarly, religion[25] and sexual orientation[26 27] have been linked to weight and body image. It is, therefore, important to understand the extent to which the current evidence base is applicable to those who are most at risk of experiencing poorer obesity-related health outcomes and have more severe obesity.

This study aimed to determine the extent to which the findings from intervention studies of weight management, as exemplified by long-term RCTs, are generalisable and applicable to those most at risk, particularly underserved groups with severe obesity. To examine these questions, we set out:

1. To describe inclusion and exclusion criteria for RCTs of adult weight management interventions, and in those trials:
2. To describe efforts to tailor recruitment strategies to improve recruitment of people from underserved groups.
3. To describe efforts to culturally adapt interventions to increase the accessibility or appeal to shared characteristics of an underserved group.
4. To describe reported baseline characteristics and outcomes for these groups.

## METHODS

Our data set comprised 131 RCTs included in a systematic review of weight management interventions for adults with BMI ≥35 kg/m², as part of the REBALANCE Project (REview of Behaviour And Lifestyle interventions for severe obesity: AN evidenCE synthesis; NIHR HTA 15/09/04).[10] BMI ≥35 kg/m² was chosen as this is a cut-off often used for accessing bariatric surgery or secondary care weight management clinical services in the UK. Eligible interventions included diet (including, but not limited to, very low-calorie diets and meal replacements), lifestyle (including combination of diet, physical activity and types of counselling), bariatric surgery or orlistat.

RCTs were restricted to publications after 1990 up to May 2017 to reflect more recent clinical practice. Literature searching was conducted in June 2016 and updated in April/May 2017. Details of the literature search method and search strategy are available in the online supplemental files. Trials had to report long-term data on weight change (≥1 year of follow-up) and include trial populations with a baseline mean or median BMI ≥35 kg/m². The decision to focus on long-term RCTs for this study was informed by the preference for high-quality, long-term evidence of lasting effectiveness in guideline documents[8 28–31] and are, therefore, most likely to influence treatment policy decisions. Reports published as abstracts or conference proceedings only were excluded. Three reviewers screened titles, abstracts and full-text reports with a 10% quality assessment check. We attempted to contact the first, second and last authors of the main publications to identify all additional materials (ie, protocols, trial materials and diet books) to inform our data extraction for the main REBALANCE report. Full details of the completed REBALANCE Project, including the protocol, have been published.[10]

In the absence of definitions of underserved groups, we identified underserved groups by using protected characteristics informed by the PROGRESS (Place of residence, Race/ethnicity/culture/language, Occupation, Gender/sex, Religion, Education, Socioeconomic status, Social capital)-Plus framework[32] and the UK Equality Act 2010.[33] Four reviewers (MA-M, MC, MI and CR) conducted double data coding of each RCT for their reporting of whether trials reported details of their inclusion and exclusion criteria, trial recruitment strategies, baseline characteristics and outcome reporting for the following characteristic groups, with disagreements resolved by consensus:

► Older age
► Physical health
► Mental health (including, but not limited to, depression, psychosis, schizophrenia, substance abuse and eating disorders)
► Comorbidities (eg, types 1 and 2 diabetes mellitus)
► Gender/sex (including RCTs recruiting only men or women)
► Sexual orientation
► Gender reassignment
► Marriage or civil partnership status
► Pregnancy
► Religion or belief
► Place of residence/housing (including residents of supported accommodation and homeowner status)
► Race or ethnicity
► Language
► Occupational status
► Education/literacy
► SES, including individual SES and participants recruited from rural or disadvantaged geographical locations
► Social capital (including social support networks and/or social isolation)

- ► PROGRESS-Plus (personal characteristics associated with discrimination (eg, age, disability), features of relationships (eg, smoking parents, excluded from school), time-dependent relationships (eg, leaving the hospital, respite care, other instances where a person may be temporarily at a disadvantage))

For inclusion and exclusion criteria, trials were coded by predefined categories indicating whether any of the characteristic groups were clearly reported in the inclusion/exclusion criteria, or, where details were not reported, whether the setting of the trial encouraged/discouraged inclusion of individuals from a particular characteristic group (eg, recruitment was set in a health centre predominantly serving people from a characteristic group), or by whether it was unclear that the trial included/excluded people from any of the characteristic groups. For baseline and outcome reporting, we coded whether the protected characteristic was reported and, if reported, whether it was reported for individual treatment groups or the trial population as a whole. Where subgroup analyses were reported, we coded these according to whether it was clear/unclear from the study report that analyses were preplanned. For a trial to be coded as having adapted their recruitment strategy for an underserved group, additional efforts to employ strategies that would appeal to that particular group (eg, held recruitment days in particular settings or developed recruitment materials in multiple languages) had to be demonstrated. Trials that solely recruited from one characteristic group using conventional recruitment methods (eg, newspaper or radio advertisements) were not coded as having an adapted recruitment strategy. Similarly, trials had to demonstrate that their interventions were designed with an underserved group in mind to be coded as having delivered a culturally adapted intervention. The focus of this study was to provide a description of trial methods and trial reporting to answer each of our research questions in relation to underserved groups; therefore, no formal statistical analysis was conducted.

### Patient and public involvement

Although we did not consult patient representatives for this particular analysis, three patient representatives were members of the REBALANCE Project Advisory Group, who contributed to developing the research questions, data interpretation and reporting of the research findings.

### RESULTS

From the total of 131 included trials, 19 were identified from database searching and 112 were identified from autoalert searching. The Preferred Reporting Items for Systematic Reviews and Meta-Analyses flow chart and list of included studies are presented in the online supplemental files. Of the 131 trials, 41 (31.3%) provided us with additional materials for their publications for the main REBALANCE report, although most of the information for the current analysis was obtained from the primary publications. The majority (81 of 131, 61.8%) of included studies were set in North America (80 in the USA and 1 in the USA and Canada), 41 out of 131 (31.3%) were in European countries (including 8 in the UK), 8 (6.1%) were in the Southern Hemisphere (6 in Australia, 1 in New Zealand and 1 in Australia and New Zealand) and 1 in Brazil. None of the trials were set in low-income countries. Just under half (62 of 131, 47.3%) of the studies were published between 2011 and 2017. Five (3.8%) trials were linked to publications reporting qualitative data.[34–38] Few trials had follow-up duration longer than 12 months, with the exception of the US Look AHEAD trial[39] (median duration of 9.6 years), and four trials with follow-up times of 5 years.[40–43] The Look AHEAD trial was the largest trial, including over 5000 participants. Interventions were wide ranging, including very low calorie (19 of 131, 14.5%), orlistat (12 of 131, 9.2%), bariatric surgery (11 of 131, 8.4%) and other lifestyle weight management programmes incorporating diet and physical activity advice (89 of 131, 67.9%). Details of the characteristics of the included studies can be found in the online supplemental files of the REBALANCE report.[10]

### Trial recruitment

More than half of the trials (71 of 131, 54%) recruited participants either solely or partially through a health service provider, for example, either solely from outpatient clinics and general practices,[44 45] or by physician referral and targeted mailing.[46] Recruitment methods were unclear or not reported in 14 trials (10.7%).[47–59] Recruitment methods for the other trials were mainly advertisements in local newspapers or other media. Based on their reporting, only three (2.3%) trials were judged to have adapted their recruitment strategies to appeal to underserved groups.[36 60 61] These preplanned strategies included holding pre-recruitment presentations in US schools,[60] recruitment events at football stadiums of Scottish Premier League football clubs[36] and having bilingual staff take informed consent and provide written consent forms in both English and Spanish languages.[61] It was unclear in a further five (3.8%) trials whether recruitment strategies had been adapted beyond conventional methods.[53–55 62 63]

Regarding adaptions to interventions, seven trials (5.3%, six from the USA and one from New Zealand)[61 63–68] recruited participants from diverse racial or ethnic groups and reported cultural adaptations to their interventions. Five (3.8%) of these trials[63–67] included advice on regional or culturally adapted recipes and foods for specific ethnic groups. Two trials (1.5%)[61 68] had interventions that were delivered by bilingual staff. One trial[68] also reported that the intervention was designed for delivery in populations with limited literacy and numeracy and impaired access to health-promoting resources. While two trials[52 69] recruited participants from workplace settings (an automobile manufacturer and a university; both were not considered to meet the PROGRESS-Plus occupation definition), and

**Table 1** The number (and per cent) of trials (n=131) reporting inclusion and exclusion criteria by protected characteristics included in the REBALANCE systematic review of RCTs

| | Inclusion | | | Exclusion |
|---|---|---|---|---|
| | **Protected characteristic is reported in the inclusion criteria, n (%)** | **Protected characteristic is not reported in inclusion criteria, but an effort was made to recruit from the protected characteristic group, n (%)** | **Unclear if the protected characteristic was targeted for inclusion or if the trial unintentionally recruited solely/mainly from the protected characteristic group, n (%)** | **Protected characteristic is reported in the exclusion criteria, or the reported inclusion criteria clearly excluded the protected characteristic, n (%)** |
| Place of residence/housing | 5 (3.8) | 0 | 0 | 0 |
| Race/ethnicity | 7 (5.3) | 5 (3.8) | 2 (1.5) | 1 (0.8) |
| Occupation status | 0 | 0 | 0 | 0 |
| Women only | 19 (14.5) | 0 | 4 (3.1) | 0 |
| Pregnancy | 0 | 0 | 0 | 72 (54.9) |
| Men only | 4 (3.0) | 0 | 1 (0.76) | 5 (3.8) |
| Religion/belief | 1 (0.8) | 0 | 0 | 0 |
| Education/literacy | 0 | 0 | 0 | 0 |
| Socioeconomic status | 3 (2.3) | 3 (2.3) | 0 | 0 |
| Marital status | 1 (0.8) | 0 | 0 | 1 (0. 8) |
| Older age | 2 (1.5) | 1 (0.8) | 0 | 82 (62.6)* |
| Physical health | 10 (7.6) | 0 | 0 | 51 (38.9) |
| Diabetes type 1 | 0 | 0 | 0 | 15 (11.5) |
| Diabetes type 2 | 28 (21.4) | 0 | 0 | 29 (22.1) |
| Diabetes (type 1 and 2 or type not reported) | 0 | 0 | 0 | 3 (2.3) |
| Mental health | 6 (4.6) | 0 | 0 | 76 (58.0) |
| Substance abuse | 0 | 0 | 0 | 58 (44.2) |
| Eating disorder | 0 | 0 | 0 | 35 (26.7) |
| Language | 0 | 0 | 0 | 16 (12.2) |

*Includes eight RCTs recruiting participants up to 75 years, one RCT recruited participants up to 76 years and three RCTs recruited participants aged up to 80 years.
RCTs, randomised controlled trials; REBALANCE, REview of Behaviour And Lifestyle interventions for severe obesity: AN evidenCE synthesis.

one trial recruited married participants,[70] the trials did not report any attempts to alter their recruitment strategies or interventions to appeal to underserved occupation or social capital groups. The trials did not report recruitment strategies or adaptions to interventions for sexual orientation.

A further five trials (all from the USA)[71–75] sought recruitment from specific racial or ethnic groups and reported intervention adaptations to increase cultural salience. Four of these trials[71–74] included culturally specific dietary advice and recipes. One trial described including bilingual interventions[71] and three trials described including interventionists from specific racial or ethnic groups.[73–75] Two trials[73 74] described using logos and programme identification 'for African-Americans', with one of these trials[73] including a video greeting from an African-American principal investigator.

The number of trials reporting inclusion and exclusion criteria by protected characteristic groups is presented in table 1. Four older trials (3.1%)[47 52 76 77] did not report any inclusion criteria, and inclusion criteria were

unclear in one further study.[42] Seven (5.3%)[52 70 78–82] trials did not report any explicit exclusion criteria and did not report that they had no exclusion criteria. Eight (6.1%) trials reported either full[36 38 45 72 83 84] or partial[60 85] justification for their exclusion criteria. Justification for exclusion criteria included prevention of poor adherence and losses to follow-up,[38 45 84 85] such as substance abuse, mental health problems or cognitive impairment (that might, in the opinion of the investigators, hinder participation), lower BMI cut-offs for Asian people,[60] non-English-language speakers where the intervention required English language comprehension,[84] influence of pregnancy and breast feeding on weight,[84] taking medications that influence weight,[72] and contraindications or safety concerns associated with participating in the intervention (eg, risk of participants with cardiovascular disease participating in exercise programmes).[36 45 84 85] Over half (58.0%) of the trials reported excluding people with mental health conditions and 44.2% excluded people with substance abuse or addiction issues. The majority of trials also excluded

**Table 2** The number (and per cent) of trials (n=131) reporting protected characteristics at baseline in the REBALANCE systematic review of RCTs

|  | Protected characteristic is reported at baseline for each intervention arm | Protected characteristic is reported at baseline for the whole trial | Total |
| --- | --- | --- | --- |
| Age | 126 (96.2%) | 4 (3.0%) | 130 (99.2%) |
| Physical health | 10 (7.6%) | 0 | 10 (7.6%) |
| Mental health | 10 (7.6%) | 0 | 10 (7.6%) |
| Diabetes | 6 (4.6%) | 0 | 6 (4.6%) |
| Sex | 126 (96.2%) | 2 (1.5%) | 128 (97.7%) |
| Gender reassignment | 0 | 0 | 0 |
| Sexual orientation | 0 | 0 | 0 |
| Marriage/civil partnership status | 38 (29.0%) | 0 | 38 (29.0%) |
| Pregnancy | 0 | 0 | 0 |
| Place of residence/housing | 6 (4.6%) | 0 | 6 (4.6%) |
| Occupation status | 27 (20.6%) | 1 (0.8%) | 28 (21.4%) |
| Education/literacy | 51 (38.9%) | 2 (1.5%) | 53 (40.5%) |
| Socioeconomic status | 29 (22.1%) | 1 (0.8%) | 30 (22.9%) |
| Social capital | 2 (1.5%) | 0 | 2 (1.5%) |
| Religion/belief | 1 (0.8%) | 0 | 1 (0.8%) |
| Race/ethnicity | 74 (56.5%) | 2 (1.5%) | 76 (58.0%) |
| PROGRESS-Plus | 2 (1.5%) | 0 | 2 (1.5%) |

PROGRESS, Place of residence, Race/ethnicity/culture/language, Occupation, Gender/sex, Religion, Education, Socioeconomic status, Social capital; RCTs, randomised controlled trials; REBALANCE, REview of Behaviour And Lifestyle interventions for severe obesity: AN evidenCE synthesis.

adults from older age groups and based on current or planned pregnancy.

Twenty-one (16.0%) trials were judged to have inclusion criteria that might have implicitly excluded certain disadvantaged groups, such as people who do not have healthcare insurance,[69 86] people who do not belong to a particular religious community group,[87] people without regular internet[35] or telephone[88] access, and English language comprehension.[44 45 49 56 60 68 89–96] In a further 40 (30.5%) trials, it was unclear if trial recruitment could have implicitly excluded disadvantaged groups. Few trials reported their inclusion criteria so as to include particular underserved groups or were judged to have made efforts to maximise trial recruitment from these groups. When trial recruitment was targeted, this was usually to recruit women only (19 trials) or people with type 2 diabetes (28 trials).

### Reporting of baseline characteristics

Details of the number of trials reporting baseline characteristics of their participants by each of the protected characteristic groups are presented in table 2. The majority of trials reported age (99.2%) and sex (97.7%) in their description of baseline participant characteristics. Just over half of the trials (58.0%) reported race or ethnicity. Education history was less well reported (40.5%). SES was reported by 22.9%, and occupation status by 21.4%. Of the trials that were not specifically for people with diabetes, six (4.6%) included diabetes in their reporting of baseline characteristics. Two (1.5%) trials reported whether

people lived alone or not (coded as social capital).[97 98] Few trials reported details of the other protected characteristics, and none reported details of gender reassignment, sexual orientation or pregnancy.

### Outcome reporting

Details of the number of trials reporting outcomes by each of the protected characteristics are shown in table 3. Very few trials reported outcomes by protected characteristic groups. Where outcomes were reported by protected characteristics, the most common group was sex (47.3%), followed by race or ethnicity (16.8%).

## DISCUSSION

### Main findings

Our findings demonstrate that most trialists testing weight management strategies to help adults with severe obesity fail to consider populations who are most at risk of poorer health outcomes. Almost all trials were from high-income countries, where lower SES and income are associated with a greater prevalence of obesity, particularly severe obesity.[1–5] Few trials reported adapting recruitment to appeal to underserved groups or reported culturally adapting their trial materials for ethnic groups or people with limited English language literacy or numeracy. This is concerning as limiting the accessibility or appeal of trials could limit the representativeness of the trial population, and thus limit the generalisability of trial findings. Only 6.1% of trials gave any justification for their exclusion

**Table 3** The number (and per cent) of trials (n=131) reporting outcome data for protected characteristics in the REBALANCE systematic review of RCTs

| | Trial recruitment was targeted at people from the protected characteristic group | One or more outcome(s) reported for the protected characteristic in planned subgroup analysis | One or more outcome(s) reported for the protected characteristic—unclear if subgroup analysis was preplanned | Total |
|---|---|---|---|---|
| Older age | 2 (1.5%)* | 2 (1.5%) | 3 (2.3%) | 7 (5.3%) |
| Physical health | 10 (7.6%) | 0 | 0 | 10 (7.6%) |
| Mental health | 6 (4.6%) | 0 | 2 (1.5%) | 8 (6.1%) |
| Diabetes | 28 (21.3%) | 0 | 1 (0.8%) | 29 (22.1%) |
| Sex | 23 (17.5%)† | 17 (13.0%) | 22 (16.8%) | 62 (47.3%) |
| Gender reassignment | 0 | 0 | 0 | 0 |
| Sexual orientation | 0 | 0 | 0 | 0 |
| Marriage/civil partnership status | 1 (0.8%) | 1 (0.8%) | 3 (2.3%) | 5 (3.8%) |
| Pregnancy | 0 | 0 | 0 | 0 |
| Place of residence/housing | 0 | 2 (1.5%) | 0 | 2 (1.5%) |
| Occupation status | 2 (1.5%) | 2 (1.5%) | 4 (3.1%) | 8 (6.1%) |
| Education/literacy | 0 | 1 (0.8%) | 6 (4.6%) | 7 (5.3%) |
| Socioeconomic status | 5 (3.8%) | 4 (3.1%) | 3 (2.3%) | 12 (9.2%) |
| Social capital | 1 (0.8%) | 3 (2.3%) | 2 (1.5%) | 6 (4.6%) |
| Religion/belief | 1 (0.8%) | 0 | 0 | 1 (0.8%) |
| Race/ethnicity | 8 (6.1%) | 5 (3.8%) | 9 (6.9%) | 22 (16.8%) |
| PROGRESS-Plus | 0 | 0 | 0 | 0 |

*Both trials recruited participants aged ≥65 years.
†Nineteen women-only trials, four men-only trials.
PROGRESS, Place of residence, Race/ethnicity/culture/language, Occupation, Gender/sex, Religion, Education, Socioeconomic status, Social capital; RCTs, randomised controlled trials; REBALANCE, REview of Behaviour And Lifestyle interventions for severe obesity: AN evidenCE synthesis.

criteria, yet more than half excluded participation for age or mental health reasons. Where justification for exclusion was reported, the rationale included excluding people who were deemed likely to have poor intervention adherence or were more likely to be lost to follow-up, such as people with substance abuse, cognitive impairment or mental health problems. Excluding these groups could lead to an unrealistic estimation of the real-world effectiveness of interventions. Just over half of the trials reported participants' race or ethnicity, and only around one-fifth reported SES. Where outcomes were reported for underserved groups, the most common analysis was by sex (47.3%), followed by race or ethnicity (16.8%); however, where analyses were presented as subgroups, it was often unclear whether these analyses were planned or unplanned. Similarly, some smaller trials might have been underpowered to detect differences in treatment effects between subgroups, but this was also unclear from trial reporting. This finding was also demonstrated by Liu and colleagues,[99] who highlighted a lack of transparent reporting of intentions to analyse race and ethnicity subgroups in Cochrane intervention reviews. Only 3.1% of the trials we reviewed reported outcomes according to SES. Few trials reported outcomes by the other protected characteristics.

Although we were limited by the available information in the published reports, our findings are concerning. In almost all trials, it is difficult to assess the generalisability of findings to the wider population of adults with severe obesity. There is clear evidence[1–5 11 12] that underserved groups with lower incomes, less education, lower SES, intellectual and physical disabilities, or poorer mental health are more at risk of obesity, particularly severe obesity, in high-income countries, especially the USA which provided the majority of trials examined. We do not have relevant data to be able to comment on the reasons for poor reporting. Nevertheless, the lack of reporting for characteristics reflecting underserved groups suggests that trial investigators did not consider or faced barriers that prevented their inclusion in the design, recruitment, and analysis or reporting of their interventions.

Our finding that few trials adapted their recruitment methods or interventions to appeal to underserved groups suggests lack of engagement with underserved people with obesity in the design of services. This is important, given that a systematic review of qualitative research by Sutcliffe and colleagues[100] showed how service users have perspectives that should inform weight management services to improve their reach. From systematic reviews, researchers have clearly demonstrated the need

to involve communities in all stages of research in order to enhance the engagement and generalisability of that research, acknowledging that this requires extended time frames and greater costs.[21 101] For example, Ni She and colleagues[102] undertook a rapid realist review of the mechanisms and resources needed to engage underserved, seldom-heard groups in health and social care research, with items grouped by an expert panel under the headings of environmental and social planning, service provision, guidelines, fiscal measures, communication and marketing, and regulation and legislation. In the USA, Arnegard and colleagues[103] have also called on the NIH's stakeholder groups to redouble their efforts to encourage sex/gender-aware reporting of biomedical investigations. We endorse this call following the findings from our previous systematic review of weight management interventions for men with obesity.[8] Our review highlighted the paucity of evidence for men, who are less likely to take part in weight management interventions, and the lack of engagement of men in all aspects of intervention design, and optimal trial recruitment processes of weight management.[8]

While the reasons for the under-representation of underserved groups in RCTs are likely to be complex and multifaceted, with many known and unknown barriers to participation, there is evidence that, for some groups, willingness to participate is not a predominant factor.[104] Mindful of the need to improve the engagement of underserved groups in research in the UK, the NIHR set up the INCLUDE Project,[19] which has led to the INCLUDE ethnicity framework[105]; providing four key questions on who should be involved in research, and how to facilitate involvement. Others have investigated and found a lack of external validity in trials for people with asthma,[106] type 2 diabetes[107 108] and neurological disease,[109] or a failure to justify exclusion criteria in trials of cardiovascular disease prevention[110] other than for safety reasons. These publications did not consider the SES or educational attainment of trial participants. In a systematic review of 305 trials of clinical conditions, He and colleagues[111] found high exclusion rates in trials for people with hypertension (83.0%), lipid-lowering drugs in primary prevention (85.9%), type 2 diabetes (81.7%), chronic obstructive pulmonary disorder (COPD) (84.3%) and asthma (96.0%), with no strong evidence that exclusion rates had changed with time.

More recently, others have also highlighted the small number of intervention studies testing out weight management for underserved adults,[112] or policies to assist socioeconomically disadvantaged groups.[113] A 2015 systematic review of interventions aimed at reducing socioeconomic inequalities for adults with obesity[114] found that primary care-delivered tailored weight loss programmes and group weight loss interventions had the most evidence of potential effectiveness in reducing obesity, at least in the short term, among low-income women, but there were few individual-level intervention studies and a lack of long-term evidence of effectiveness.

## Strengths and limitations

We used categories informed by the PROGRESS-Plus framework[32] and the UK Equality Act 2010[33] as the key characteristics for identifying those underserved participants who should be considered for study design, public–patient involvement, recruitment, analysis and reporting, not just for trials of weight management, but trials generally. O'Neill and colleagues[32] have shown how the prior PROGRESS framework can be used as an equity lens for systematic reviews and methodological studies; however, NIHR's INCLUDE Project has recently published a more extensive list of categories of underserved groups to consider with regard to representation in trials.[115]

Our literature search attempted to identify all long-term randomised trials published since 1990 for adults with BMI $\geq$35 kg/m$^2$ irrespective of the type of lifestyle intervention, including comparisons with orlistat and bariatric surgery. Although we included publications in any language from any country, we cannot exclude the possibility that we failed to find some trials, particularly those from low-income countries, which might not be listed in the databases we searched.

While we originally contacted authors for all available additional materials relating to our main research question for the REBALANCE Project,[10] we did not recontact authors to obtain additional information relating to the current analysis. We were also limited by poor reporting by trial authors. Some trials were statistically underpowered to detect subgroup differences, and this might explain under-reporting of underserved characteristics; however, this was unclear from the trial reports. It is also possible that some trialists were unable to obtain relevant baseline data for some underserved groups if this was deemed sensitive by an ethics committee, for example, sexual orientation. Nevertheless, we consider that most characteristics are pivotal to interpreting these trials into real-world guidance and services, so we would expect their presentation in trial publications, especially at baseline. We assessed long-term RCT evidence because it is most likely to inform guidance on weight management.[28–31] Other study designs, interventions with shorter follow-up and unpublished studies may have been more inclusive in their designs and reporting.

## Recommendations for research

Trialists should improve reporting of their justification of inclusion and exclusion criteria to meet current Consolidated Standards of Reporting Trials (CONSORT) statement guidelines,[116] and report sufficient data to allow comparisons between their populations and the populations for whom the interventions apply. Including core criteria for baseline reporting within the CONSORT checklist[116] could help to improve the completeness of reporting of these factors. NIHR's INCLUDE Project's ethnicity framework provides important factors to consider with regard to ethnic groups (https://www.trial-forge.org/trial-forge-centre/include/), which can help provide transferable considerations for other underserved

groups. However, a wider equity lens may be needed in the face of groups with multiple disadvantages. Although guidance for research will aid considerations of equity, we do not yet have ways of assessing when proportional representation in larger trials and subgroup reporting for underserved groups is insufficient. This research should be explicitly conducted with and for these underserved groups, ensuring user involvement in all stages of the research process.

## CONCLUSIONS AND RECOMMENDATIONS FOR PRACTICE

Long-term RCTs of weight management in people with BMI $\geq 35\,\text{kg/mg}^2$ have inadequate representation of and engagement with underserved groups, who are particularly relevant for health and social care services. Thus, guidance for weight management research on how to improve the representation of underserved groups in clinical trials may improve the appropriateness of that research and help inform greater engagement of underserved communities with health and social care services.

**Acknowledgements** We thank the members of the REBALANCE Project and Advisory Groups for their contributions to the REBALANCE Project. We thank Shaun Treweek and Heidi Gardner, Health Services Research Unit, University of Aberdeen, for helpful discussions on trial generalisability and inclusion of underserved groups.

**Contributors** CR, MA-M, MC, MI and AA contributed to conception and design, acquisition of data, or analysis and interpretation of data, and revised data critically for important intellectual content of this manuscript. All authors critically reviewed the manuscript and approved the final version submitted for publication. CR takes responsibility for the integrity of the data and the accuracy of the data analysis. CR is the guarantor and accepts full responsibility for the finished work and/or the conduct of the study, had access to the data, and controlled the decision to publish.

**Funding** The REBALANCE Study was funded by the National Institute for Health Research (NIHR) Health Technology Assessment Programme (project number: 15/09/04).

**Disclaimer** The views expressed are those of the author(s) and not necessarily those of the NHS, the NIHR or the Department of Health. The Health Services Research Unit is funded by the Chief Scientist Office of the Scottish Government Health and Social Care Directorates.

**Competing interests** None declared.

**Patient and public involvement** Patients and/or the public were not involved in the design, or conduct, or reporting, or dissemination plans of this research.

**Patient consent for publication** Not required.

**Ethics approval** Ethical approval was not required.

**Provenance and peer review** Not commissioned; externally peer reviewed.

**Data availability statement** Data are available upon reasonable request. All data relevant to the study are included in the article, uploaded as supplemental information, or are available from the NHIR journals library ((REBALANCE) REview of Behaviour And Lifestyle interventions for severe obesity: AN evidenCE syntheis ( nihr.ac.uk))

**ORCID iDs**
Clare Robertson http://orcid.org/0000-0001-6019-6795
Magaly Aceves-Martins http://orcid.org/0000-0002-9441-142X
Moira Cruickshank http://orcid.org/0000-0002-5182-884X
Mari Imamura http://orcid.org/0000-0003-4871-0354
Alison Avenell http://orcid.org/0000-0003-4813-5628

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
