## [Reviewer comments · BMJ Open]

ARTICLE DETAILS

TITLE (PROVISIONAL)	Does weight management research for adults with severe obesity represent them? Analysis of systematic review data.
AUTHORS	Robertson, Clare; Aceves-Martins, Magaly; Cruickshank, Moira; Imamura, Mari; Avenell, Alison

VERSION 1 – REVIEW

REVIEWER	Jull, Andrew The University of Auckland, School of Nursing
REVIEW RETURNED	23-Jul-2021

GENERAL COMMENTS	Thank you for the opportunity to review this paper. It is informative and well written. I have only minor comments: 1. The statement justifying selection of RCTs with 12 months or longer follow up (page 2 line 10 and page 20 line 31) assert that these types of trials are most likely to inform guidelines and practice. While such a claim can be made in the abstract without citation, it may be overstating the actuality without a reference. While some guidelines may limit themselves to such evidence, others might limit themselves to RCTs with 6 months or longer follow up.2. Just using the USA and UK as examples of (developed) countries that have racial or ethnic groups at higher risk of obesity ignores the evidence from Australia, New Zealand, and mainland Europe. It would be more inclusive to cite more than just UK and US examples.3. It would be useful to briefly describe the search strategy rather than just reference the REBALANCE project on page 6, particularly as the PRISMA diagram provided seems to indicate 19 studies were added to previous searches. Or a fuller description could be added as a supplement in order that the search can be replicated.4. On page 6 the authors note they tried to contact the first, second, and last authors for trial details to inform the data extraction (with 41 providing additional details). Yet it is stated in the limitations on page 20 that "we did not contact authors for additional information extracted for this analysis ...". These two statements seem contradictory and require clarifying.5. Some percentages in tables 1 and 2 have been truncated at one decimal place and some have been rounded to one decimal place. It would seem more appropriate to use rounding throughout rather than a mix of truncation and rounding.6. The finding that very few trials reported subgroup analyses will most likely be driven by the fact that trialists are dissuaded from
--

	performing subgroup analyses unless they are specified a priori (and, ideally, properly powered). The authors do raise the lack of clarity in the trials about whether reporting subgroups was planned or unplanned, but do not address the issue that lack of subgroup analyses in their findings can also be driven by good reporting practice. 6. Sentence 2 on page 20 line 22 in the limitations does not follow on from the point raised in sentence 1 and needs further clarification if it is to make sense. 7. Perhaps the most important guidance in the Recommendations is unstated - the authors have discussed the lack of user involvement in trial design and conduct on page 18, which surely should lead to the recommendation that in the future those affected by the research should be involved in the development of that research. 8. There was an update of the PRISMA Checklist published in 2020 that includes extra items compared to the 2009 checklist. Choice of checklist may have been driven by the journal rather than the authors, in which case BMJ Open needs to ensure updated checklists are provided to authors.
--	--

REVIEWER	Salis, Amanda The University of Sydney, Faculty of Medicine
REVIEW RETURNED	03-Aug-2021

GENERAL COMMENTS	This is an excellent paper that is well written, easy to understand, with an important, clear and practical take home message. May 2017 as the latest date for inclusion of publications seems a bit old. Can this be updated? In the methods, it is stated that only clinical trials with adults with severe obesity were included. What about clinical trials where adults with and without severe obesity were included? For example, clinical trials where inclusion criteria was a body mass index of 30 to 40. Would this clinical trial be excluded or included? More clarity around this would be helpful. The paragraph commencing at line 13 on page 7 would benefit from some greater explanation, and possibly an example of how this coding was done.
--

REVIEWER	Lawlor, Emma MRC Epidemiology Unit, University of Cambridge
REVIEW RETURNED	10-Aug-2021

GENERAL COMMENTS	This is a well-written paper investigating the representativeness of participants with obesity in weight management RCTs. Overall, I found it an interesting topic and found it very clear to follow. My main issue is that it refers often to another report instead of providing details of the methods and findings in the text – I feel this reduces its readability, ability to be a standalone paper and many interesting details are missing. I also feel there should be some more reference to other groups of people (e.g. Progress plus criteria, social capital, sexual orientation), as the focus is mostly on socio-economic status, ethnicity, language etc, yet this project aims to look at these other under-served groups, so feel they could be acknowledged more
--

	throughout the text. I have used the page numbers that the authors submitted Abstract Only small comments. It says 'only' a lot in the results section which is quite repetitive - it would be better saying 'a few' or referring to number of studies as only seems quite subjective and is not that descriptive. In line about 'Just over half...' it would be good to give the n in brackets or a % Page 5 I feel that the first paragraph maybe doesn't set the scene as well as it could and is maybe a bit of issue with flow as it seems like bit of a list –maybe another sentence to introduce the topic is need (maybe after the first sentence) Line 10 – should it be 'greater risk' than 'risks'? Line 38: This could be more personal preference, but should it be 'is a major public health...'? Line 54: Reference for this statement starting on this line would be great Is there any more justification for why severe obesity? As surely the points you've made is also relevant to people with obesity too Page 6 Line 41: Checking other literature, people have defined severe obesity as BMI over 40 – can you justify with a sentence in the text why you went with 35 (apart from being part of the larger project)? Line 44: Does lifestyle not include diet? And what about physical activity? Any ones on meal replacement food/drinks? Line 46: Is there a reference or anything more concrete to justify why 1990 is used as cut off point specifically? Could there be a little more detail on both inclusion and exclusion criteria? Page 7 Line 16: Full stop after framework unnecessary Line 29: Are you not interested in other adaptations made, rather than just cultural? Page 8
--	--

Line 52: PPI – do you mean they contributed to the project overall, or this specific review? It would be lovely if there were any additional detail how they influenced the review such as outlined in GRIPP 2 checklist

I really think there should be more detail here on screening, databases and analysis etc. I understand it is published elsewhere as part of a main review paper, but I feel this paper should be able to stand alone a bit better. Just a few sentences, but it should be addressed. Or if you're really struggling for word count, some detail could go in the appendix.

Relevant to my last point, the sentence beginning on line 47 could maybe be given a sub-heading on analysis, and just elaborated a bit more

Page 9

Results

Line 12: I find starting with this as a first line a little odd as I wouldn't think that the number providing additional details is a huge take away finding. Can you say a bit more about how many were found from database searches etc?

Can you have a PRISMA diagram, or at least refer to it here rather than in the methods?

Line 26: You've said 'only five' – to me it sounds a little odd as this wasn't something you were going to extract or are particularly interested in.

Again, I feel that the results could be so much more illuminating for the reader if there was a bit more detail on the studies, so they can image the types of trials this reporting is taking place in

Could you even add n or % after where you describe the interventions on line 36

Line 49: 'partially through a health service...' – what were these other complimentary methods e.g. was this other method to try to attract other groups?

Throughout results, I think there could be some more referencing after you mention the number of studies

Could you maybe say for each of your main sections if you got this information from the papers, or had to seek it out (e.g. not readily available to the standard reader/clinician). I think this could also help to unpick more which characteristics etc are more readily reported

Page 10

Line 5: Meant to be stadium?

Line 24: Are these 2 trials from the 5 you mentioned before? I find it difficult to know due to absence of referencing earlier

I find it difficult to see clearly how each of these paragraphs are saying something different/why they aren't combined – could the first

sentence of each paragraph made it explicitly clear what it is about to present?

I wondered if trials were able to, or how they could adapt recruitment strategies to appeal to groups such as sexual orientation, social capital, occupational status etc? Could you add something about these types of under-represented groups, or if there was no findings say that to ensure they are also included?

I had wondered if any reported adjusting their recruitment strategy to include these groups during the trial on recognition of lack of under-served groups so far, or if they had started/planned this type of recruitment all along.

Page 11

Line 5: I would like to know more about the reasons for exclusion. For example, was there clinical reasons e.g. it would be unsafe for pregnant women to undergo bariatric surgery – or was it more that the trial just decided to not to recruit them. Just to get a sense of what they have done in the trial is actually ethically or clinically sound, rather than that they could impact the results detrimentally/didn't want them

Page 12

In the table, can you have n and (%) in the top column?

Can you also include the progress plus criteria, even if it is zero

I wondered if there was any details about how the characteristics information was actually collected? For example, if recruiting through healthcare service provider (which half your studies did) maybe the medical records did not record some of the information or it was more on a need to know basis so it's not really the fault of the researchers. Further, some information they may not have been allowed to collect by ethics committees etc as it was not seen as essential to the study aim (e.g. sexual orientation) or would have any impact on findings. Further, some of these questions could have been really off putting or seen as too sensitive by participants which researchers would avoid as they are trying to have high recruitment rates. Could there be some comments on this in the discussion?

Page 17

Discussion

I feel that there should be more acknowledgement in the discussion of reasons why this poor reporting may have happened and the practical barriers that people doing trials may have (similar to my previous comment) – this could also then help with some recommendations for practice.

You've focused a lot more in introduction and discussion about income, disability etc – is there any ways that sexual orientation, language, religion influence weight? Or why reporting that is

	important and could have an impact? Better reporting over time as progress plus and checklists have emerged? Line 17 onwards: I feel a bit that the % etc in the first discussion paragraph is a bit more just repeating the result Line 47: I'm not quite clear what this line is about Page 18 Line 5: I feel this line is a little too strong - there could have been barriers etc Line 15: I feel this sentence makes it sound as if you extracted this type of outcome data (which I don't think you did from the methods) from your included trials (if you did, please reference them) – could this maybe be better placed as a recommendation for practice/future research section? Page 20 Line 3: Seems odd where limitations section is – I'd have it at the end of the discussion – or is the paragraph starting at line 40 meant to be framed as a strength? As it seems a little odd where it currently is if it's not about strengths Page 21 I feel the conclusion could be strengthened, and a lot more could be said about recommendations for practice. Page 22 If there was 131 included studies, why are there only 95 references? Or it could be good to have them in an appendix Page 35 Why the databases picked up so little studies that you actually included? Could that be mentioned in the limitations section and acknowledged? Page 36 Do you want to use the more up to date PRISMA checklist from 2020? Or maybe the 2009 is what is recommended by BMJ Open Thank you for letting me review this paper – I am looking forward to seeing the revisions and good luck with them. Hopefully my
--	--

	comments have been helpful.
--	-----------------------------

VERSION 1 – AUTHOR RESPONSE

Reviewer: 1

Prof. Andrew Jull, The University of Auckland, The University of Auckland

Comments to the Author:

Thank you for the opportunity to review this paper. It is informative and well written. I have only minor comments:

Thank you for your comments.

1. The statement justifying selection of RCTs with 12 months or longer follow up (page 2 line 10 and page 20 line 31) assert that these types of trials are most likely to inform guidelines and practice. While such a claim can be made in the abstract without citation, it may be overstating the actuality without a reference. While some guidelines may limit themselves to such evidence, others might limit themselves to RCTs with 6 months or longer follow up.

Thank you for bringing this to our attention. We have now added references for the statements in the methods section “The decision to focus on long-term RCTs for this study was informed by the preference for high-quality, long-term evidence of lasting effectiveness in guideline documents “ and in the discussion section “We assessed long-term RCT evidence because it is most likely to inform guidance on weight management” to support this claim. We have restructured the abstract and removed mention of evidence to inform guideline documents.

2. Just using the USA and UK as examples of (developed) countries that have racial or ethnic groups at higher risk of obesity ignores the evidence from Australia, New Zealand, and mainland Europe. It would be more inclusive to cite more than just UK and US examples.

We agree that there is evidence of higher obesity risk for some racial and ethnic groups beyond the UK and US. We have added the examples of Australia and New Zealand with supporting citations to better illustrate this. We also believe the first sentence of the background alerts the reader to this being an issue in multiple countries.

3. It would be useful to briefly describe the search strategy rather than just reference the REBALANCE project on page 6, particularly as the PRISMA diagram provided seems to indicate 19 studies were added to previous searches. Or a fuller description could be added as a supplement in order that the search can be replicated.

We have now added these details to Appendix 1, which describes the literature search methods and search strategy.

4. On page 6 the authors note they tried to contact the first, second, and last authors for trial details to inform the data extraction (with 41 providing additional details). Yet it is stated in the limitations on page 20 that “we did not contact authors for additional information extracted for this analysis ...”.

These two statements seem contradictory and require clarifying.

Thanks for alerting us to this point. We agree the current wording causes confusion for the reader. When we contacted authors we asked for additional materials and information to guide our data extraction for the main review questions for the REBALANCE HTA report. We did not contact the authors again to ask questions specifically relating to the questions addressed in the secondary analysis presented in this manuscript. We have amended the text in the manuscript, as detailed below, to improve clarity on this point.

Methods

“We attempted to contact the first, second and last authors of the main publications to identify all additional materials (i.e. protocols, trial materials and diet books) to inform our data extraction for the main report.

Discussion

“We did not re-contact authors for the additional information extracted for this analysis and were limited by poor reporting by trial authors.”

5. Some percentages in tables 1 and 2 have been truncated at one decimal place and some have been rounded to one decimal place. It would seem more appropriate to use rounding throughout rather than a mix of truncation and rounding.

Thanks for highlighting this. We have now rounded all percentages in Tables, 1, 2 and 3 to one decimal place.

6. The finding that very few trials reported subgroup analyses will most likely be driven by the fact that trialists are dissuaded from performing subgroup analyses unless they are specified a priori (and, ideally, properly powered). The authors do raise the lack of clarity in the trials about whether reporting subgroups was planned or unplanned, but do not address the issue that lack of subgroup analyses in their findings can also be driven by good reporting practice.

We agree that it is not good practice to perform subgroup analyses if the study lacks adequate statistical power. We believe that it would be helpful for authors to report this detail in relation to under-served groups and that it is always useful to present the characteristics of under-served groups at baseline. We have amended the text in the discussion to clarify our point

“We did not re-contact authors for the additional information extracted for this analysis and were limited by poor reporting by trial authors. Some trials were statistically underpowered to detect subgroup differences, and this might explain under-reporting of under-served characteristics; however, this was unclear from the trial reports. It is also possible that some trialists were unable to obtain relevant baseline data for some under-served groups if this was deemed sensitive by an ethics committee, e.g. sexual orientation. Nevertheless, we consider these characteristics to be pivotal to interpreting these trials into real-world guidance and services, so would expect their presentation in trial publications, especially at baseline.”

6. Sentence 2 on page 20 line 22 in the limitations does not follow on from the point raised in sentence 1 and needs further clarification if it is to make sense.

Thank you for bringing this to our attention. This sentence has now been amended and moved to the results, as detailed below

“Where outcomes were reported for under-served groups, the most common analysis was by sex (47.3%) followed by race or ethnicity (16.8%); however, where analyses were presented as subgroups, it was often unclear whether these analyses were planned or unplanned. Similarly, some smaller trials might have been underpowered to detect differences in treatment effects between subgroups, but this was also unclear from trial reporting. This finding was also demonstrated by Liu and colleagues (2020), who highlighted a lack of transparent reporting of intentions to analyse race and ethnicity subgroups in Cochrane intervention reviews.”

7. Perhaps the most important guidance in the Recommendations is unstated - the authors have discussed the lack of user involvement in trial design and conduct on page 18, which surely should lead to the recommendation that in the future those affected by the research should be involved in the development of that research.

We agree this is an important recommendation. We have amended the final sentence in the recommendations, as detailed below, to better illustrate this issue

“This research should be explicitly conducted with and for these under-served groups, ensuring user involvement in all stages of the research process.”

8. There was an update of the PRISMA Checklist published in 2020 that includes extra items compared to the 2009 checklist. Choice of checklist may have been driven by the journal rather than the authors, in which case BMJ Open needs to ensure updated checklists are provided to authors. Thank you. We have now provided the PRISMA 2020 checklist.

Reviewer: 2

Prof. Amanda Salis, The University of Sydney

Comments to the Author:

This is an excellent paper that is well written, easy to understand, with an important, clear and practical take home message.

Thank you for your comments.

May 2017 as the latest date for inclusion of publications seems a bit old. Can this be updated? Ideally, we would like to update the search but, unfortunately, we do not have the resources to do this. There are approximately 17,223 potentially relevant titles and abstracts to screen in Medline since 2018.

In the methods, it is stated that only clinical trials with adults with severe obesity were included. What about clinical trials where adults with and without severe obesity were included? For example, clinical trials where inclusion criteria was a body mass index of 30 to 40. Would this clinical trial be excluded or included? More clarity around this would be helpful.

Thank you for highlighting this. We have now amended text in the methods section, as detailed below, to clarify that trials of BMI 30 to 40 would have been eligible for inclusion if the mean/median BMI of the trial population was BMI $>35\text{kg/m}^2$ at baseline.

“Trials had to report long-term data on weight change (≥ 1 year of follow-up) and include trial populations with a baseline mean or median BMI $\geq 35\text{kg/m}^2$.”

The paragraph commencing at line 13 on page 7 would benefit from some greater explanation, and possibly an example of how this coding was done.

We have revised the text in this section and expanded our description of the data coding as detailed below

“In the absence of definitions of under-served groups, we identified under-served groups by using protected characteristics informed by the PROGRESS-Plus framework and the UK Equality Act 2010. Four reviewers (MA-M, MC, MI and CR) conducted double data coding of each RCT for their reporting of whether trials reported details of their inclusion and exclusion criteria, trial recruitment strategies, baseline characteristics, and outcome reporting for the following characteristic groups, with disagreements resolved by consensus:

- Older age
- Physical health
- Mental health (including, but not limited to, depression, psychosis, schizophrenia, substance abuse and eating disorders)
- Comorbidities (e.g. Types 1 and 2 diabetes mellitus)
- Gender/sex (including RCTs recruiting only men or women)
- Sexual orientation
- Gender reassignment
- Marriage or civil partnership status
- Pregnancy
- Religion or belief
- Place of residence/housing (including residents of supported accommodation and homeowner status)
- Race or ethnicity
- Language

- Occupational status
- Education/literacy
- Socioeconomic status (SES), including individual SES and participants recruited from rural or disadvantaged geographical locations
- Social capital (including social support networks and/or social isolation)
- PROGRESS Plus (personal characteristics associated with discrimination (e.g. age, disability), features of relationships (e.g. smoking parents, excluded from school), time-dependent relationships (e.g. leaving the hospital, respite care, other instances where a person may be temporarily at a disadvantage)

For inclusion and exclusion criteria, trials were coded by pre-defined categories indicating whether any of the characteristic groups were clearly reported in the inclusion/exclusion criteria, or, where details were not reported, whether the setting of the trial encouraged/discouraged inclusion of individuals from a particular characteristic group (e.g. recruitment was set in a health centre predominantly serving people from a characteristic group), or by whether it was unclear that the trial included/excluded people from any of the characteristic groups. For baseline and outcome reporting, we coded whether the protected characteristic was reported and, if reported, whether it was reported for individual treatment groups or the trial population as a whole. Where subgroup analyses were reported, we coded these according to whether it was clear/unclear from the study report that analyses were pre-planned. For a trial to be coded as having adapted their recruitment strategy for an under-served group, additional efforts to employ strategies that would appeal to that particular group (e.g. held recruitment days in particular settings or developed recruitment materials in multiple languages) had to be demonstrated. Trials that solely recruited from one characteristic group using conventional recruitment methods (e.g. newspaper or radio advertisements) were not coded as having an adapted recruitment strategy. Similarly, trials had to demonstrate that their interventions were designed with an under-served group in mind to be coded as having delivered a culturally adapted intervention. The focus of this study was to provide a description of trial methods and trial reporting to answer each of our research questions in relation to under-served groups; therefore, no formal statistical analysis was conducted. "

Reviewer: 3

Dr. Emma Lawlor, MRC Epidemiology Unit

Comments to the Author:

This is a well-written paper investigating the representativeness of participants with obesity in weight management RCTs. Overall, I found it an interesting topic and found it very clear to follow. My main issue is that it refers often to another report instead of providing details of the methods and findings in the text – I feel this reduces its readability, ability to be a standalone paper and many interesting details are missing. I also feel there should be some more reference to other groups of people (e.g. Progress plus criteria, social capital, sexual orientation), as the focus is mostly on socio-economic status, ethnicity, language etc, yet this project aims to look at these other under-served groups, so feel they could be acknowledged more throughout the text.

Thank you for your comments. We are constrained by the article word limit, however, we have included substantial additional details to improve readability, whilst keeping the focus of the manuscript on the current analysis. We have been clear that few trials reported details for these other under-served groups in the manuscript. We have now included text clarifying that two trials were coded as reporting social capital at baseline. We have also added text to the first paragraph in the discussion section to highlight that these groups were not reported by the trials.

I have included my full review in the word document I have uploaded to the system.

Thank you for taking the time to provide these helpful comments. We have incorporated your suggestions where we felt that this was possible/appropriate. We have provided our responses to your comments below.

Additional comments from Reviewer 3

Abstract

Only small comments. It says 'only' a lot in the results section which is quite repetitive - it would be better saying 'a few' or referring to number of studies as only seems quite subjective and is not that descriptive.

Thank you for highlighting this. We have amended the text in the abstract

In line about 'Just over half...' it would be good to give the n in brackets or a %
We have now added the percentage to this sentence.

Page 5

I feel that the first paragraph maybe doesn't set the scene as well as it could and is maybe a bit of issue with flow as it seems like bit of a list –maybe another sentence to introduce the topic is need (maybe after the first sentence)

We have added an additional sentence to this paragraph. We hope this improves the readability.

Line 10 – should it be 'greater risk' than 'risks'?

Thank you. We have amended to greater risk.

Line 38: This could be more personal preference, but should it be 'is a major public health...'?

Thank you for this suggestion. We have now amended this sentence to read “ a major public health challenge”

Line 54: Reference for this statement starting on this line would be great

We have now added references for this statement.

Is there any more justification for why severe obesity? As surely the points you've made is also relevant to people with obesity too

We have now explained why the BMI ≥ 35 kg/m² cut-off was chosen for the main review and in keeping with clinical practice. The sentence in the methods section reads as follows

BMI ≥ 35 kg/m² was chosen as this is a cut-off often used for accessing bariatric surgery or clinical services in the UK.

Page 6

Line 41: Checking other literature, people have defined severe obesity as BMI over 40 – can you justify with a sentence in the text why you went with 35 (apart from being part of the larger project)?
Please see the above response.

Line 44: Does lifestyle not include diet? And what about physical activity? Any ones on meal replacement food/drinks?

We have added additional text to clarify these points, as detailed below

“Eligible interventions included diet (including, but not limited to, very low calorie diets and meal replacements), lifestyle (including combinations of diet, physical activity, and types of counselling), bariatric surgery, or orlistat.”

Line 46: Is there a reference or anything more concrete to justify why 1990 is used as cut off point specifically? Could there be a little more detail on both inclusion and exclusion criteria?

The 1990 cut off date was chosen to reflect more recent clinical practice. We have included justification for the 1990 cut off date in this sentence

“RCTs were restricted to publications after 1990 up to May 2017 to reflect more recent clinical practice.”

Additional details for inclusion and exclusion criteria have been added.

Page 7

Line 16: Full stop after framework unnecessary

Thank you. We have corrected this typo.

Line 29: Are you not interested in other adaptations made, rather than just cultural?

We have adopted the broad sense of 'cultural' to refer to the shared characteristics of a group of people. Any adaptation that aimed to increase the appeal or accessibility of the intervention would have been coded as culturally adapted. We have amended the text to clarify this

3. to describe efforts to culturally adapt interventions to increase the accessibility or appeal to shared characteristics of an under-served group,

Page 8

Line 52: PPI – do you mean they contributed to the project overall, or this specific review? It would be lovely if there were any additional detail how they influenced the review such as outlined in GRIPP 2 checklist

We think it is clear from the current wording that we did not consult our patient and public partners for this analysis. They were consulted and contributed to the main HTA review as detailed in the current manuscript. We feel that providing additional detail in this paper could mislead the reviewer into falsely believing that the PPI relates to the analysis described in this manuscript rather than the original review.

I really think there should be more detail here on screening, databases and analysis etc. I understand it is published elsewhere as part of a main review paper, but I feel this paper should be able to stand alone a bit better. Just a few sentences, but it should be addressed. Or if you're really struggling for word count, some detail could go in the appendix.

To keep within the article word limit, we have included additional text on database searching in Appendix 1. The following sentences have also been added to the methods section.

"Literature searching was conducted in June 2016 and updated in April/May 2017. Details of the literature search method and search strategy are available in Appendix 1 of the supplementary files."

"Reports published as abstracts or conference proceedings only were excluded. Three reviewers screened titles, abstracts and full text reports with a 10% quality assessment check."

Relevant to my last point, the sentence beginning on line 47 could maybe be given a sub-heading on analysis, and just elaborated a bit more

We are not sure what elaboration is requested. However, we have now included substantial additional details describing our methods and believe that this is sufficient.

Page 9

Results

Line 12: I find starting with this as a first line a little odd as I wouldn't think that the number providing additional details is a huge take away finding. Can you say a bit more about how many were found from database searches etc?

We have now added the following sentence to the first line of the results section.

"From the total of 131 included trials, 19 were identified from database searching and 112 were identified from autoalert searching. "

Can you have a PRISMA diagram, or at least refer to it here rather than in the methods?

The PRISMA diagram was supplied as supplementary material. We have referred to this in the results section.

Line 26: You've said 'only five' – to me it sounds a little odd as this wasn't something you were going to extract or are particularly interested in.

We have now amended this sentence to remove the word 'only'

Again, I feel that the results could be so much more illuminating for the reader if there was a bit more detail on the studies, so they can image the types of trials this reporting is taking place in
We have provided additional details and hope this improves readability. We are, however, constrained by the article word limit. We also believe that providing lengthy descriptions on the intervention components for the trials distracts from the focus of the current analysis. We have instead chosen to direct interested readers to the study characteristics table provided in the supplementary files of the main REBALANCE report.

Could you even add n or % after where you describe the interventions on line 36

We have now provided this information.

Line 49: 'partially through a health service...' – what were these other complimentary methods e.g. was this other method to try to attract other groups?

We have added the following text to this sentence to provide examples

More than half of the trials (71/131, 54%) recruited participants either solely or partially through a health service provider, for example, either solely from outpatient clinics and general practices or by physician referral and targeted mailing

We have also corrected a typo for the number of trials where recruitment methods were unclear or not reported.

Throughout results, I think there could be some more referencing after you mention the number of studies

We have now provided additional references throughout the results section.

Could you maybe say for each of your main sections if you got this information from the papers, or had to seek it out (e.g. not readily available to the standard reader/clinician). I think this could also help to unpick more which characteristics etc are more readily reported

Most of the information obtained in the supplementary materials related to information around the components of the interventions (frequency of contact, behavioural change components, nutritional components, etc) rather than data relating to under-served groups. We have amended the following sentence in the results section to clarify that most of the information for the analysis described in this manuscript was obtained from the primary publications.

Of the 131 included trials, 41 (31.3%) provided us with additional materials for their publications, although most of the information for the current analysis was obtained from the primary publications.

Page 10

Line 5: Meant to be stadium?

We have used the Latin plural stadia, but have now amended to stadiums to better align with common usage and to improve readability and understanding.

Line 24: Are these 2 trials from the 5 you mentioned before? I find it difficult to know due to absence of referencing earlier

We have now added citations for the seven studies to the earlier sentence.

I find it difficult to see clearly how each of these paragraphs are saying something different/why they aren't combined – could the first sentence of each paragraph made it explicitly clear what it is about to present?

We have now added the following text to the second paragraph to indicate the emphasis is on

interventions rather than recruitment strategies in this paragraph.
“Regarding adaptations to interventions ...”

I wondered if trials were able to, or how they could adapt recruitment strategies to appeal to groups such as sexual orientation, social capital, occupational status etc? Could you add something about these types of under-represented groups, or if there was no findings say that to ensure they are also included?

We have now included the following text with supporting citations

“While two trials recruited participants from workplace settings (an automobile manufacturer and a university; both were not considered to meet the PROGRESS-Plus occupation definition), and one trial recruited married participants, the trials did not report any attempts to alter their recruitment strategies or interventions to appeal to under-served occupation or social capital groups. The trials did not report recruitment strategies or adaptations to interventions for sexual orientation.”

I had wondered if any reported adjusting their recruitment strategy to include these groups during the trial on recognition of lack of under-served groups so far, or if they had started/planned this type of recruitment all along.

We have added text to the following sentence to indicate that these strategies were pre-planned

“These pre-planned strategies included holding pre-recruitment presentations in US schools, recruitment events at football stadiums of Scottish Premier League football clubs and having bilingual staff take informed consent and provide written consent forms in both English and Spanish languages.”

Page 11

Line 5: I would like to know more about the reasons for exclusion. For example, was there clinical reasons e.g. it would be unsafe for pregnant women to undergo bariatric surgery – or was it more that the trial just decided to not to recruit them. Just to get a sense of what they have done in the trial is actually ethically or clinically sound, rather than that they could impact the results detrimentally/didn't want them

We were limited by the number of trials reporting justification for their exclusion criteria. We have now reported examples of some of the rationale for exclusion from the limited number of trials that reported this information. Please see the amended text below

“Justification for exclusion criteria included prevention of poor adherence and losses to follow-up, such as substance abuse, mental health problems, or cognitive impairment that might, in the opinion of the investigators, hinder participation, lower BMI cut-offs for Asian people, non-English language speakers where the intervention required English language comprehension, influence of pregnancy and breastfeeding on weight, taking medications that influence weight, and contraindications or safety concerns associated with participating in the intervention (e.g. risk of participants with cardiovascular disease participating in exercise programmes). “

Page 12

In the table, can you have n and (%) in the top column? Can you also include the progress plus criteria, even if it is zero

We have amended Table 1 to include n (%) in the top columns. We have included additional details of the progress plus criteria and amended the ordering of information to better reflect the PROGRESS-plus acronym. As detailed in our methods, our data extraction was informed by PROGRESS-plus criteria and, therefore, some variable names may not exactly correspond to the PROGRESS-plus labels. For example, we have reported 'marital status' in this Table instead of 'social capital' as we feel this is more informative for the reader.

I wondered if there was any details about how the characteristics information was actually collected? For example, if recruiting through healthcare service provider (which half your studies did) maybe the

medical records did not record some of the information or it was more on a need to know basis so it's not really the fault of the researchers. Further, some information they may not have been allowed to collect by ethics committees etc as it was not seen as essential to the study aim (e.g. sexual orientation) or would have any impact on findings. Further, some of these questions could have been really off putting or seen as too sensitive by participants which researchers would avoid as they are trying to have high recruitment rates. Could there be some comments on this in the discussion? While we accept that some characteristics of under-served might be deemed potentially ethically sensitive, we believe that researchers would be able to obtain data on most under-served groups by including questionnaire items in baseline demographic questionnaires. We have added additional text to the discussion as follows

"It is also possible that some trialists were unable to obtain relevant baseline data for some under-served groups if this was deemed sensitive by an ethics committee, e.g. sexual orientation. Nevertheless, we consider that most characteristics are pivotal to interpreting these trials into real-world guidance and services, so we would expect their presentation in trial publications, especially at baseline.

Page 17

Discussion

I feel that there should be more acknowledgement in the discussion of reasons why this poor reporting may have happened and the practical barriers that people doing trials may have (similar to my previous comment) – this could also then help with some recommendations for practice. Unfortunately, we do not have data to confirm the reasons for poor reporting. We have added the following sentence to the discussion, and also included recommendations for CONSORT in the recommendations for research section.

"We do not have relevant data to be able to comment on the reasons for poor reporting."

"Including core criteria for baseline reporting within the CONSORT checklist could help to improve the completeness of reporting of these factors."

You've focused a lot more in introduction and discussion about income, disability etc – is there any ways that sexual orientation, language, religion influence weight? Or why reporting that is important and could have an impact?

We have added the following sentence with supporting references to the introduction

"Similarly, religion and sexual orientation have been linked to weight and body image."

We also have also included text in the discussion section to indicate why excluding people with limited English language or failing to adapt trial materials for people with limited English language is problematic for representation and generalisability.

Better reporting over time as progress plus and checklists have emerged?

It would be interesting to explore this in future work but, unfortunately, we do not have the resources to carry out formal analysis on reporting over time for this manuscript.

Line 17 onwards: I feel a bit that the % etc in the first discussion paragraph is a bit more just repeating the result

We have now amended this section to better highlight our discussion points.

Line 47: I'm not quite clear what this line is about

We have now simplified this sentence as follows

"Although we were limited by the available information in the published reports trial reporting and the information contained in additional materials obtained from around a third of trials, our findings are concerning."

Page 18

Line 5: I feel this line is a little too strong - there could have been barriers etc

We have amended this sentence as follows

“Nevertheless, the lack of reporting for characteristics reflecting under-served groups suggests that trial investigators did not consider, or faced barriers that prevented their inclusion in the design, recruitment, analysis or reporting of their interventions.”

Line 15: I feel this sentence makes it sound as if you extracted this type of outcome data (which I don't think you did from the methods) from your included trials (if you did, please reference them) – could this maybe be better placed as a recommendation for practice/future research section?

We have now modified this sentence as follows

“Our finding that few trials appeared to have adapted their recruitment methods or interventions to appeal to under-served groups suggests lack of engagement with under-served people with obesity in the design of services, which is important, given that a systematic review of qualitative research by Sutcliffe and colleagues showed how service users have perspectives that should inform weight management services to improve their reach.”

Page 20

Line 3: Seems odd where limitations section is – I'd have it at the end of the discussion – or is the paragraph starting at line 40 meant to be framed as a strength? As it seems a little odd where it currently is if it's not about strengths

We have re-ordered this section under the sub-heading 'strengths and limitations'.

Page 21

I feel the conclusion could be strengthened, and a lot more could be said about recommendations for practice.

We believe our conclusion section is informative; however, we have expanded this section. The text now reads as follows

“Long-term RCTs of weight management in people with BMI >35mg2 lack generalisability and engagement with under-served groups, who are particularly relevant for health and social care services. Thus, guidance for weight management research on how to improve the representation of under-served groups in clinical trials may improve the appropriateness of that research and help inform greater engagement of under-served communities with health and social care services.”

Page 22

If there was 131 included studies, why are there only 95 references? Or it could be good to have them in an appendix

We have now included a list of included studies as an Appendix and refer to this in the methods section

Page 35

Why the databases picked up so little studies that you actually included? Could that be mentioned in the limitations section and acknowledged?

We have now included additional details on the literature searching in Appendix 1 that explain this point.

Page 36

Do you want to use the more up to date PRISMA checklist from 2020? Or maybe the 2009 is what is recommended by BMJ Open

Thank you. The PRISMA 2020 checklist is now provided.

VERSION 2 – REVIEW

REVIEWER	Jull, Andrew The University of Auckland, School of Nursing
REVIEW RETURNED	10-Feb-2022

GENERAL COMMENTS	Thank you for the opportunity to review this paper. It presents a detailed analysis of the lack of inclusivity in trials addressing long term management of overweight and obesity. The methods for obtaining the studies are detailed and the analysis is appropriate. It is a useful examination of the literature and adds to the body of evidence showing that certain populations are poorly served by the research community. I wish to raise three quite minor points: Firstly, it is certainly fair to say that the listed trials from 1990 to 2017 are less than representative of populations most at risk of overweight and obesity and these populations are under-served by the research. The conclusion in the abstract is certainly accurate, but the conclusion in the paper seems less judicious when it states there is a "lack of generalisability". Certainly, the findings may have less applicability to some populations than others, but they are still generalisable although with less certainty that the effects will be similar to those observed in the trial. To state that the body of evidence lacks generalisability overstates the issue in my view as it may imply that the studies are not generalisable at all, whereas in reality most trials may be less generalisable to at risk populations, but it is not that they cannot be generalised. I know it is a nuanced point, and perhaps I am too cautious about making categorical calls, but the current conclusion does seem less judicious than that in the abstract. Secondly, line 22 on page 10 states 41 trials provided the authors with additional materials for their publications, but lines 12-14 on page 23 state "we did not re-contact authors for additional information...". These two statements could be read as contradictory and clarification would be helpful. Thirdly, and I acknowledge it is an absolute triviality, but there is a repeated comma on line 10 page 20. Thanks once again for the chance to read this work and congratulations to the authors for such a detailed examination of the issue.
---

VERSION 2 – AUTHOR RESPONSE

Reviewer: 1

Prof. Andrew Jull, The University of Auckland, The University of Auckland

Comments to the Author:

Thank you for the opportunity to review this paper. It presents a detailed analysis of the lack of inclusivity in trials addressing long term management of overweight and obesity. The methods for obtaining the studies are detailed and the analysis is appropriate. It is a useful examination of the literature and adds to the body of evidence showing that certain populations are poorly served by the research community.

Thank you for taking the time to review our revised manuscript.

I wish to raise three quite minor points:

Firstly, it is certainly fair to say that the listed trials from 1990 to 2017 are less than representative of populations most at risk of overweight and obesity and these populations are under-served by the research. The conclusion in the abstract is certainly accurate, but the conclusion in the paper seems less judicious when it states there is a "lack of generalisability". Certainly, the findings may have less applicability to some populations than others, but they are still generalisable although with less certainty that the effects will be similar to those observed in the trial. To state that the body of evidence lacks generalisability overstates the issue in my view as it may imply that the studies are not generalisable at all, whereas in reality most trials may be less generalisable to at risk populations, but it is not that they cannot be generalised. I know it is a nuanced point, and perhaps I am too cautious about making categorical calls, but the current conclusion does seem less judicious than that in the abstract.

Thank you for raising this important issue. We agree and have amended the following sentences:

Discussion section, page 20

"In almost all trials, it is difficult to assess ~~would be impossible to assess~~ the generalisability of findings to the wider population of adults with severe obesity.

Conclusions section, page 24

"Long-term RCTs of weight management in people with BMI ≥ 35 mg² ~~lack generalisability~~ have inadequate representation of and engagement with under-served groups, who are particularly relevant for health and social care services.

Secondly, line 22 on page 10 states 41 trials provided the authors with additional materials for their publications, but lines 12-14 on page 23 state "we did not re-contact authors for additional information...". These two statements could be read as contradictory and clarification would be helpful.

We apologise for the confusing statements and have now amended the following sentences to provide clarification:

Methods section, page 7

"We attempted to contact the first, second and last authors of the main publications to identify all additional materials (i.e. protocols, trial materials and diet books) to inform our data extraction for the main REBALANCE report. Full details of the completed REBALANCE project, including the protocol, have been published."

Results section, page 10

“Of the 131 trials, 41 (31.3%) provided us with additional materials for their publications for the main REBALANCE report, although most of the information for the current analysis was obtained from the primary publications.

Discussion section, page 23

While we originally contacted authors for all available additional materials relating to our main research question for the REBALANCE project,¹⁰ we did not re-contact authors ~~for to obtain~~ the additional information ~~extracted for relating to the is current analysis.~~ ~~and~~We were also limited by poor reporting by trial authors.

Thirdly, and I acknowledge it is an absolute triviality, but there is a repeated comma on line 10 page 20.

Thank you for spotting this typo. We have deleted the extra comma.